# A Design for Wayfinding: Developing a Mobile Application to Enhance Spatial Orientation at Taipei Metro

**Kuang-Ting Huang * and Meng-Yan Zhou**

Department of Architecture, National Taipei University of Technology, Taipei 10608, Taiwan;
t109528011@ntut.org.tw
* Correspondence: kthuang@ntut.edu.tw; Tel.: +886-2-2771-2171 (ext. 2927)

**Abstract:** Taipei Metro, since its inception in 1996, has become the most important public transport option for commuters and travelers in the metropolitan Taipei area, delivering over two million daily rides. Nevertheless, the interior environment of Taipei Metro has a reputation for being disorienting, especially to the infrequent passengers. By incorporating the methods of behavioral mapping and visibility analysis, this study argues that the occurrence of disorientation is highly dependent on visual properties of Taipei Metro's interior layout. Specifically, the number of decision-making stops and the visibility conditions of stairs and escalators are found to be particularly influential. To enhance the passengers' wayfinding experience, a mobile application comprised of two components is proposed. The Route Planner is to advise the passengers to avoid the areas that cause disorientation, while the Navigator, by providing the panoramic views of certain locations, can help the passengers reach their destinations more easily.

**Keywords:** wayfinding; behavioral pattern; visibility; space syntax





## 1. Introduction

The Taipei Metro is a metro system serving the Taipei metropolitan area, and encompasses an area of approximately 2500 km$^2$, which has a population of over seven million. Since its inception in 1996, the network of Taipei Metro has quickly expanded from 12 stations to 131, and from 10.5 km to 152.9 km [1]. In the same time frame, its daily ridership has also experienced a similar growth from forty thousand to over two million, making Taipei Metro the 22nd busiest metro system in the world [2].

Nevertheless, Taipei Metro has a reputation for being disorienting and confusing, especially to the infrequent passengers. According to a survey on its service quality conducted in 2020 by Taipei Rapid Transit Corporation, the control of passenger flow and the spatial disorientation are, respectively, rated as both the top and the second most in need of improvement by 38.7% and 33.5% of respondents [3]. As further detailed in the survey report, the most commonly identified cause of feeling disoriented is the complexity of the floor plan configuration and the lack of wayfinding guidance. Specifically, most of Taipei Metro stations have multiple underground levels that consist of a basement concourse and several boarding platforms (Figure 1). Inevitably, the vertical travel between different levels increases the difficulty for the passengers to find their way to either the exit gates or to specific destinations inside the station.

To minimize the spatial disorientation in Taipei Metro stations, this paper firstly reviews the concept of spatial orientation and presents a case study of Zhongxiao Xinsheng Station using the methods of behavioral mapping and visibility analysis. Based on the data obtained from the case study, a mobile application that will enhance the wayfinding experience of passengers is proposed.

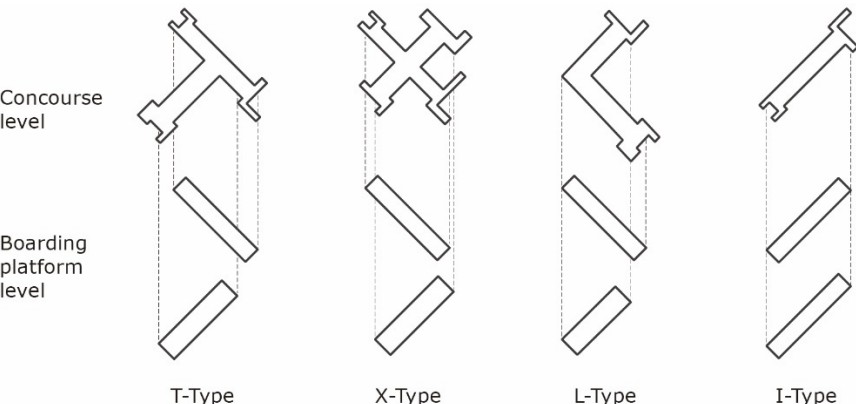

**Figure 1.** Typical floor plans of Taipei Metro stations.

## 2. Spatial Orientation and Wayfinding in Metro Station

Spatial orientation has long been an important issue in the field of environmental psychology. Since the 1980s, there is a growing scholarship focusing on the identification of factors associated with wayfinding behaviors [4–6]. Among all identified factors, the authors of [7], through a review of existing literature, have distinguished two broad types as follows: user factors and environmental factors (Table 1). The former is concerned with the relationship between the cognitive ability to assimilate spatial information and the navigating behaviors, while the latter is focused on the physical elements that may enhance or reduce the spatial cognition.

**Table 1.** Types of wayfinding factors.

| Type | Subtype | Element |
|:---:|:---:|:---:|
| User factors | Wayfinding cognition | Spatial memories, logical association, information pick-up, neuroanatomy, etc. |
| | Wayfinding behavior | Behavioral performance, navigation pattern |
| | Individual and group differences | Age, sex, psychological state, culture |
| Environmental factors | Environmental elements | Floor plan configuration, regions, edges, paths, nodes, landmarks |
| | Environmental cues | Signs, maps, other environmental factors |

For the type of user factors, the authors of [8] pointed out that aging is associated with the decline of spatial cognition and is a particularly important issue for studying spatial orientation. By comparing the wayfinding performance of different age groups, it is found that older adults not only demonstrate less effective spatial memory [9], and the ability to pick up environmental information [10] and establish logical association [11], they also need more directional guidance to find their way around. Regarding the environmental factors, the influence of floor plan configuration on wayfinding performance has received the most attention in published literature. Specifically, many studies suggested that in a spatially complex environment there is an observed increase of time needed to perform wayfinding tasks, the number of wrong turns, and the occurrence of backtracking [12–14].

Despite the rich literature on spatial orientation, most of it is carried out in hospitals, museums, and shopping malls. In the few studies that investigate metro stations, researchers unanimously affirm that the underground multi-level environment is the key

factor affecting the passengers' wayfinding performance [15]. The authors of [16], by examining the case of two metro stations in Brussels, specified that the main reasons are the absence of direct sunlight and the lack of recognizable spatial features, while the authors of [17] studied the case of Hong Kong, pointing out that most passengers lose their orientation during the vertical travel between different levels. The most common way to minimize the negative influence discussed above is to install directional signs and maps to help the passengers understand their orientation [18]. Nevertheless, to search for directional guidance itself may be a difficult wayfinding task, especially for the passengers with deficits in either mobility or spatial memory.

In the recent years, to overcome the issue of spatial disorientation, there is a growing interest among both software developers and academic researchers in applying mobile technology on wayfinding [19]. For example, 3D Wayfinder, developed by 3D Technologies R&D, uses short-range broadcast devices to provide customized digital signage for their users, while NaviLens by Neosistec is an enhanced kind of QR code that can be scanned to access wayfinding information [20,21]. Another example is the mobile application of Gate-to-Go proposed by [22]. It uses the reference number of platform screen doors and exit gates as the index to help the passengers locate their destinations and find the shortest routes. In sum, although the recent innovations on wayfinding technology have demonstrated the potential of mobile devices in enhancing the accessibility of directional information, not many of them have been formally tested and made available to the public. Despite the passengers' dissatisfaction about spatial disorientation, especially in the case of the Taipei Metro, there has been a paucity of empirical studies devoted to the topic, until now. In order to fill in the literature gap and propose a viable solution, the rest of this study is structured as follows: Section 3 explains the methods employed in this case study. Section 4 provides the results of the case study and discusses the relationship between the passengers' behavioral patterns and the metro station's visibility conditions. In Section 5, a mobile application called Rideway is presented as a solution to the issue of spatial disorientation. Section 6 concludes this study with a summary along with directions for future research.

## 3. Methods

To address the two types of factors reviewed above, two methods were applied in this study: behavioral mapping and visibility analysis.

### 3.1. Behavioral Mapping

Wayfinding is a decision-making process of determining a route from one location to another [23]. To understand the influence of user factors on the passengers' wayfinding experience in metro stations, the method of behavioral mapping was employed in this study. Behavioral mapping is an observational research method that can simultaneously capture the observed behaviors, as well as the information about the environmental location and context where the behaviors take place. This method can not only record the actual use of a space, but more importantly, it can also denote how the environment is supporting or influencing that use. Meanwhile, behavioral mapping is a valuable method for recording environmental behaviors, because it does not rely on the self-report of users [24]. For the same reason, it is particularly valuable for capturing the wayfinding behaviors of metro passengers, with whom methods such as interviews and questionnaires may be less effective.

In practice, the observers were required to randomly pick up a passenger, either at the entrance level or platform level, follow him/her and record his/her movements and other behaviors, such as when they checked maps, chatted with friends, took phone calls, etc. The tracing could stop, either when he/she left the metro station or boarded the train. All records were first marked with a pen on a preprinted floor plan and then were converted into an analyzable format of data. Specifically, this study used the solid line to represent the choice of the shortest route between the concourse level and the platform

level, and the dashed line to represent the actual route recorded by the observer. By making comparisons between the two types of routes, the purpose was to analyze the pattern of the passengers' movement and, in the meanwhile, to identify the locations where the spatial disorientation occurred.

### 3.2. Visibility Analysis

To understand the influence of environmental factors on the passengers' wayfinding experience, this study specifically applied the method of visibility analysis developed by space syntax. Space syntax is a body of theories and techniques that investigates the relationship between space and behavior. The core work of this theory was introduced by Bill Hillier and Julienne Hanson in 1984 [25]. Since then, it has experienced consistent development and has now become an active field, covering multiple overlapping disciplines such as: architecture, urban design, urban planning, transport, and interior design [26].

The method of visibility analysis was a space syntax method that features a quantitative analysis of visual properties in the built environment. Based on a graph representation of the gross geometry of built environment, the method could offer a comprehensive modelling of how the space may be used and perceived by its occupants [27]. In practice, the modelling method was processed using the open-source software depthMapX, developed by the Space Syntax Laboratory at the Bartlett, University College, London. As Figure 2A,B illustrated, the interior space of metro stations in the software was represented as a regular grid with units in the size of a foot step. By measuring the mutual visual relationship between every two grids, the software generated a table containing several visibility measures (depth, connectivity, integration, etc.) and, in the meanwhile, a graph (Figure 2D) with a color gradient showed the spatial distribution of visibility. The warmer colors indicated higher values, whereas the cooler colors indicated the lower values. Specifically, this study used the measure of integration as the primary indicator for assessing the visibility conditions of metro stations. The higher value of integration indicated the better permeability and accessibility.

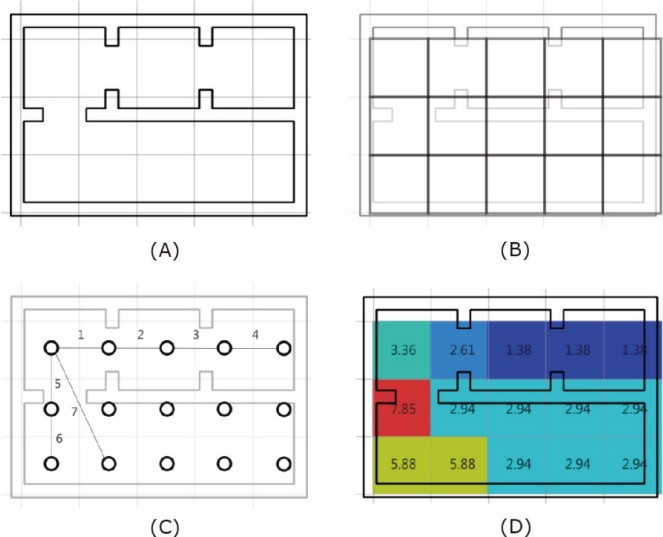

**Figure 2.** Spatial representation in visibility analysis: (**A**) floor plan, (**B**) grid, (**C**) visual relations, and (**D**) visibility graph.

## 4. Analysis

To further explain how the methods presented above can be applied to explore the issue of spatial disorientation in the Taipei Metro, this study selected Zhongxiao Xinsheng station as the site for analyzing the behavior patterns and visibility conditions of passengers. As explained in the previous section, the analysis contained two interrelated parts: behavioral mapping and visibility analysis.

For behavioral mapping, it can be observed in the case of Zhongxiao Xinsheng that although most of the passengers were taking the shortest routes, disorientation still occurred quite often and had a very regular pattern. As illustrated in Figure 3, all the mismatches between solid lines (the shortest route) and dashed lines (the actual route of passengers) were located at either stairs or escalators, which indicated that most of the disorientation occurred when moving between different levels. For example, among transfer passengers there was a common hesitation and confusion about which stairs (or escalator) to take. Especially in the case of Zhongxiao Xinsheng, because there was a multi-story atrium in the center that provided a direct visual link between the platform at the second basement, and the concourse at the first basement. As such, after many transfer passengers disembarked from the train, they would follow the crowd and proceed upstairs without noticing where the stairs (or escalator) to the other platform actually were (as seen in the dashed line C in Figure 3). As a result, they were either driven to the wrong direction or became lost in the wrong platform.

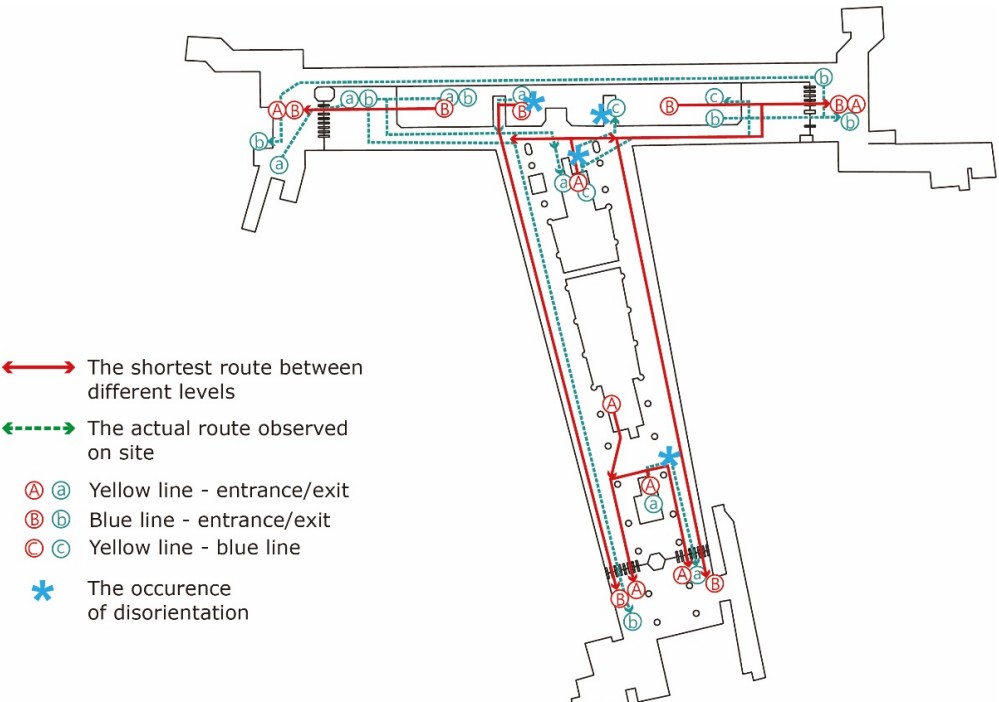

**Figure 3.** Behavioral pattern of passengers in Zhongxiao Xinsheng.

The second portion of analysis was to quantify the influence of visibility condition on the passengers' behaviors. As Figure 4 demonstrated, the average value of visual integration for the concourse level of Zhongxiao Xinsheng station is 5.78, with the highest at 9.55 and the lowest at 3.29. In Table 2, it could be further observed that the areas showing the highest visibility were located where the movement of passengers heading to different destinations intersected (with the average value of integration at 9.12; not only 57.8% higher than the average but also the second highest among all spatial divisions). To further compare the results of visibility analysis with the behavioral patterns discussed earlier, two factors that were influential on the passengers' wayfinding behaviors could be identified.

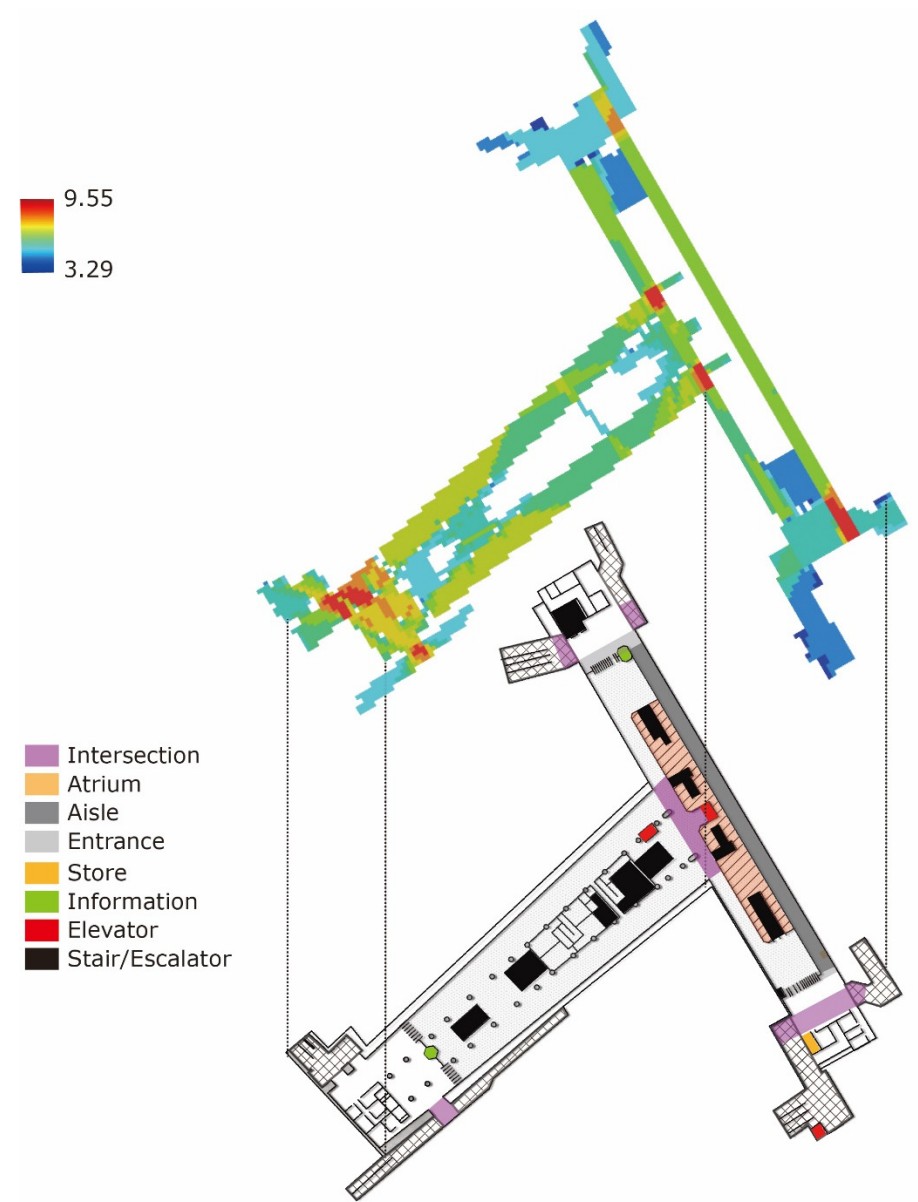

**Figure 4.** Visibility analysis of the concourse level at Zhongxiao Xinsheng station.

**Table 2.** Quantified comparison of visibility between different spatial divisions of the concourse level at Zhongxiao Xinsheng station.

| Division | Intersection | Atrium | Aisle | Entrance/Exit |
|---|---|---|---|---|
| Visual Integration | 9.12 (+57.8%) | 9.34 (+61.6%) | 3.87 (−33.0%) | 4.67 (−19.2%) |
| **Division** | **Service** | **Information** | **Elevator** | **Stair/Escalator** |
| Visual Integration | 6.04 (+4.5%) | 5.89 (+1.9%) | 7.34 (+26.9%) | 8.47 (+46.5%) |

Note: The number in brackets represent it's percent difference with the average visual integration.

First, the number of decision-making points during the passengers' wayfinding process was important in influencing their orientation. As seen in Figure 3, every point of decision making was where disorientation could occur. Despite multiple directional signs posted, often the passengers still did not have enough visual aids to receive the information for making the right decision. Secondly, the direction of stairs and escalators in relation

to the passengers' movement was another important factor. The stairs and escalators were usually located between two corridors heading in opposite directions, and therefore had higher visibility (with the visual integration at 8.47, 46.5% higher than the average). However, because most of the stairs and escalators were massive and sight-blocking, many passengers when walking through the corridors could not decide where to go, due to the limited vision.

According to the above analysis, there was an apparent corresponding relationship between the passengers' behavioral patterns and the metro station's visibility conditions. To enhance the wayfinding experience in Taipei Metro, the following section proposes a mobile application called Rideway, comprising of two functional components: Route Planner and Navigator.

### 5. Design Solution

Based on the above analysis, the design of Rideway is aimed to generate an interactive route map that can guide the users to reach wherever they want to go. Specifically, the purpose is not simply to provide the shortest route between their origins and destinations, but more importantly, the route will help the app users to avoid the areas that may cause disorientation. As demonstrated in Figure 5, once the user opens Rideway, a map of overlapped layers will be displayed on the screen. On the bottom is the floorplan of metro station, with the user's standing point represented as a red circle. On the top of the floorplan is a color gradient showing the likelihood of disorientation occurrence.

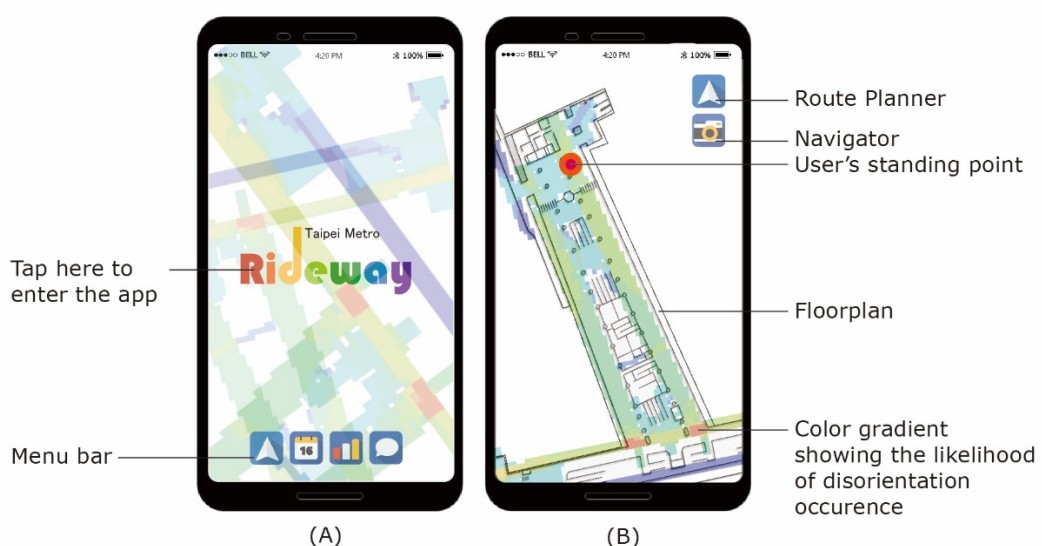

**Figure 5.** Interface of Rideway: (**A**) Home page and (**B**) Interactive map guiding the user to avoid disorientation.

For those users who already know where they are going, they can click the button on the upper-right corner and use the Route Planner (Figure 6). After specifying the destination, a recommended route between their present locations and their destinations will be automatically generated. While following the guiding route, the users will constantly receive notices reminding them of the potential disorientation (highlighted in the color purple). To obtain further directional information, the app users can use the Navigator by clicking the button next to Route Planner. The Navigator provides the users a panoramic view of the areas that they are about to walk into. In the panoramic view, the users can also find other useful information that can help them reach their destination more easily, such as floorplans, sections, directional signs, etc.

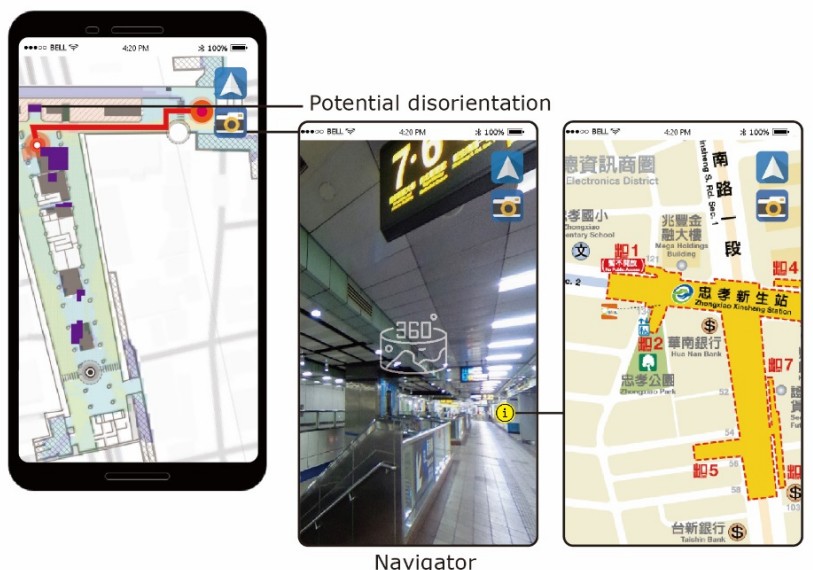

**Figure 6.** Components of Rideway for enhancing wayfinding.

In sum, the mobile application of Rideway is meant to enhance the wayfinding environment of Taipei Metro stations by providing the passengers an instant, easy access to directional guidance. Specifically, the design of Route Planner can help the passengers avoid disorientation, while the Navigator can provide panoramic views that facilitate the passengers to reach their destinations more easily. From avoiding the wrong turns and backtracking, to finding the right way, the passengers' wayfinding experience is expected to be greatly enhanced in the following three ways: 1. Reducing the time needed for finding their way to a specific destination; 2. Avoiding the unnecessary detours, especially among the transfer passengers; 3. Relieving the congestion and improving the overall efficiency of circulation during rush hour.

## 6. Conclusions

This study, by analyzing the behavioral patterns and visibility conditions of Taipei Metro passengers, has found that both user factors and environmental factors are important in causing spatial disorientation. Especially for the aging population discussed earlier, their deficits in either mobility or spatial memory can be greatly amplified in the enclosed, multi-level environment of an underground metro station. To minimize the negative influence of spatial disorientation, we designed the mobile application of Rideway. Through our study of the passengers' behavioral patterns and visibility conditions, Rideway can be easily accessed and used as a directional guidance for anyone in need. Meanwhile, the analytic methods and mobile application presented in this study can also be applied to other metro systems. Despite the differences in the spatial layout and the passengers' wayfinding habits, we need to better understand the complexity of metro stations' interior environments, and to enhance their quality of spatial orientation. In the future, more research should be conducted to refine the design of Rideway:

- The spatial typology of metro stations is of great significance in influencing the passengers' wayfinding behaviors. A comparison between different types of metro stations will reveal more details about how the environmental factors may enhance or reduce the passengers' spatial cognition.
- The current analysis of the metro stations' visibility condition is still limited. Other associated types of information, such as the location of directional signs and maps, should also be considered in the analysis, to provide a more integrated view of wayfinding environment.

**Author Contributions:** K.-T.H. was responsible for the methodology, the data analysis, original draft preparation, and final editing; M.-Y.Z. was responsible for the data collection and participated in the data analysis. All authors have read and agreed to the published version of the manuscript.

**Funding:** This research was funded by Ministry of Education, Taiwan under Grant No. PHA1090334.

**Institutional Review Board Statement:** Ethical review and approval were waived for this study since private information was not collected specifically for this study through interaction or intervention with living individuals and all subjects were de-identified during data collection and analysis.

**Informed Consent Statement:** Patient consent was waived due to the data in which there was no information identifying individuals.

**Data Availability Statement:** The data presented in this study are available on request from the corresponding author. The data is composed of two parts. The first part, the passengers' behavioral mapping, was obtained by observation. The second part, the result of visibility analysis, was compiled by the authors by using the open-source software depthMapX.

**Conflicts of Interest:** The authors declare no conflict of interest.

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
