# Peer review of "A Design for Wayfinding: Developing a Mobile Application to Enhance Spatial Orientation at Taipei Metro"

_asi, doi:10.3390/asi4040091_

Round 1
Reviewer 1 Report
Summary of the manuscript
The present manuscript examines spatial navigation in Taipei Metro with a goal of developing a mobile application that will optimize wayfinding. Specifically, the authors incorporate two important features in such mobile application: 1) the Route Planner which suggests optimal paths to avoid possible disorientation, and 2) the Navigator which help guide users on effective paths as they navigate through the Metro.
The strengths of Huang & Zhou lie in the importance and timeliness of the research question of effective and optimal wayfinding a large public transport station frequented by millions of commuters and travelers. I also appreciate the clarity with which the authors illustrate schematics associated with their approaches and results. Moreover, the reported findings could have implications for future studies investigating spatial navigation or related cognitive functions in real-world settings. Having said this, I do have some comments that I hope the authors could address.
- It is not clear what the hypothesis of the study were. In the introduction, the authors mentioned that authors mentioned that ‘there is an observed disorientation, especially for the infrequent passengers’ – could the authors expand more on this e.g., what are the sources of these data and how does ‘observed disorientation’ defined/measured here? I think this is an important point to get across as it is the main drive to the development of the mobile application. Also, it might be helpful to readers if the authors could explicitly state in the introduction what the hypotheses are in the context of the behavioral patterns and visibility conditions of passenger examined.
- Could the authors add more descriptions to the caption of figure 2, particularly to briefly point out the importance/relevance of each of the floor subplots?
- Could the authors add more descriptions to the caption of figure 5?
- I find the reported findings associated with the developed application to be intriguing, and noteworthy. I believe that the authors could expand on the impacts of such development by adding some information regarding how such device could also be useful for aging populations as well as populations with memory and cognitive deficits. This will align nicely which the ‘individual and group differences’ subtype of the ‘User factors’ already mentioned in table 1. In particular, I suggest that the authors include some short sentences and references to the discussion to state how the developed device can also benefit these populations given that past studies have shown that certain aspects of spatial cognition and navigation are intact in aging and memory-impaired individuals. Some references that will highlight these points include:
- Gazova et al (2013), Frontiers in Aging Neuroscience. Spatial navigation in young versus older adults: https://www.frontiersin.org/articles/10.3389/fnagi.2013.00094/full
- Rungratsameetaweemana & Squire (2018), Learning & Memory. Preserved capacity for scene construction and shifts in perspective after hippocampal lesions: https://www.ncbi.nlm.nih.gov/pmc/articles/PMC6049396/
Author Response
- It is not clear what the hypothesis of the study were. In the introduction, the authors mentioned that authors mentioned that ‘there is an observed disorientation, especially for the infrequent passengers’ – could the authors expand more on this e.g., what are the sources of these data and how does ‘observed disorientation’ defined/measured here? I think this is an important point to get across as it is the main drive to the development of the mobile application. Also, it might be helpful to readers if the authors could explicitly state in the introduction what the hypotheses are in the context of the behavioral patterns and visibility conditions of passenger examined.
Author response: Thank you for this important suggestion. I have modified the introduction section and added a description about the passengers’ feedback to spatial orientation in a survey conducted in 2020. Based on the same survey report, in the same paragraph I also specified “complexity of floor plan configuration” and “lack of wayfinding guidance” as the cause of feeling disoriented, and explained the purpose of this study.
- Could the authors add more descriptions to the caption of figure 2, particularly to briefly point out the importance/relevance of each of the floor subplots?
Author response: Thank you very much for pointing this out. I have modified Figure 2 and added a brief description for each floor chart under the caption. More explanation about the floor chars is given in the following paragraph. - Could the authors add more descriptions to the caption of figure 5?
Author response: Thank you for the suggestion. I have modified Figure 5 by adding more explanation on it and also added a brief description for each screenshot under the caption of Figure 5. - I find the reported findings associated with the developed application to be intriguing, and noteworthy. I believe that the authors could expand on the impacts of such development by adding some information regarding how such device could also be useful for aging populations as well as populations with memory and cognitive deficits. This will align nicely which the ‘individual and group differences’ subtype of the ‘User factors’ already mentioned in table 1. In particular, I suggest that the authors include some short sentences and references to the discussion to state how the developed device can also benefit these populations given that past studies have shown that certain aspects of spatial cognition and navigation are intact in aging and memory-impaired individuals. Some references that will highlight these points include:
Gazova et al (2013), Frontiers in Aging Neuroscience. Spatial navigation in young versus older adults: https://www.frontiersin.org/articles/10.3389/fnagi.2013.00094/full
Rungratsameetaweemana & Squire (2018), Learning & Memory. Preserved capacity for scene construction and shifts in perspective after hippocampal lesions: https://www.ncbi.nlm.nih.gov/pmc/articles/PMC6049396/
Author response: Thank you so much for your detailed explanation. The above two articles are very good reference to my study.
In order to better define the concept of spatial orientation and explain the value of the developed device, I firstly expanded the discussion about spatial orientation into an independent section. In the newly added section 2, a lot more discussion on factors associated with wayfinding behaviors is added: 1) the influence of aging on spatial memory, information pick-up, and logical association; 2) the spatial complexity (e.g. underground, multi-level environment). Also, in the conclusion section I added more explanation about the contribution of this study and specified the passengers with deficits in either mobility or spatial memory as the potential users of the developed device.
Reviewer 2 Report
This paper deals with the interesting topic to improve the Taipei metro management by proposing web application.
I recommend to be more specific in several layers, to improve the complexity of the paper. Namely, describe the developed application more precisely. Compare the methods with the other metro systems. Are such techniques applicable to any system? Discuss it in the text with focus on the limitations and potential risks. Moreover, describe and reference the figures in a textual form, as well. Finally, extend the list of references to be up-to-date, study also other systems, which are used across the world. Are you just the one dealing with the same problem? How is such problem solved in other countries, maybe?
Author Response
I sincerely appreciate the time and effort that you put forth to provide feedback on my manuscript. The following is my point-by-point response to your suggestions:
I recommend to be more specific in several layers, to improve the complexity of the paper. Namely, describe the developed application more precisely. Compare the methods with the other metro systems. Are such techniques applicable to any system? Discuss it in the text with focus on the limitations and potential risks. Moreover, describe and reference the figures in a textual form, as well. Finally, extend the list of references to be up-to-date, study also other systems, which are used across the world. Are you just the one dealing with the same problem? How is such problem solved in other countries, maybe?
Author response: Thank you so much for your recommendation and detailed explanation. In order to improve the complexity of the paper, a number of modifications are made as follows:
- In order to better define the concept of spatial orientation and explain the value of the developed application, I expanded the discussion about spatial orientation into an independent section. In the new section 2, I added two paragraphs trying to provide more details about the factors associated with wayfinding behaviors. The first paragraph is to explain in what context the disorientation may occur and what kind of influence it has, such as the increase of wayfinding time, the occurrence of backtracking. By shifting the focus to metro station, the next paragraph added two more case study references to explain how spatial disorientation actually occur and the purpose of developing a mobile application as well.
- I have modified Figure 2 and 5 by adding sub-captions under each sub-figures. Under the caption of Figure 2 and 5, a brief description for each sub-figures are also provided.
- Thank you for your suggestion about the extending the references. With the addition of section 2, I not only have added a number of new references into the manuscript; based on the newly added literature review, I also added a paragraph in the conclusion reflecting on the limitation of the study and the future direction of work.
Reviewer 3 Report
Review to asi-1430781
I am impressed by the reviewed paper – congratulations. My only concern is about the introduction section – it uses only a single reference. This section must be improved, especially using A. Natapov latest works on wayfinding issues (https://doi.org/10.1016/j.ssci.2021.105483).
I do recommend the reviewed paper for further publication.
Author Response
I am impressed by the reviewed paper – congratulations. My only concern is about the introduction section – it uses only a single reference. This section must be improved, especially using A. Natapov latest works on wayfinding issues (https://doi.org/10.1016/j.ssci.2021.105483).
I do recommend the reviewed paper for further publication.
Author response: I sincerely appreciate the time and effort that you put forth to provide feedback on my manuscript. Thank you so much for your suggestion. Natapov's latest study is a very good reference to my study.
In order to make the manuscript stronger and more complete, a number of modifications are made:
- In order to better define the concept of spatial orientation and explain the value of the developed application, I expanded the discussion about spatial orientation into an independent section. In the new section 2, I added two paragraphs trying to provide more details about the factors associated with wayfinding behaviors. The first paragraph is to explain in what context the disorientation may occur and what kind of influence it has, such as the increase of wayfinding time, the occurrence of backtracking. By shifting the focus to metro station, the next paragraph added two more case study references to explain how spatial disorientation actually occur and the purpose of developing a mobile application as well.
- With the addition of section 2, I not only have added a number of new references into the manuscript; based on the newly added literature review, I also added a paragraph in the conclusion reflecting on the limitation of the study and the future direction of work.
Round 2
Reviewer 1 Report
Thank you for your thoughtful responses to my comments. I would like to recommend the manuscript for publication.
Author Response
Thank you very much.
Reviewer 2 Report
The authors have applied some suggestions and paper looks now significantly better. However, there is still missing relevant comparison with other systems.
Compare the methods with the other metro systems. Are such techniques applicable to any system? Discuss it in the text with focus on the limitations and potential risks.
Author Response
I sincerely appreciate the time and effort that you put forth to provide feedback on my manuscript. Thank you very much for this important suggestion. I have added a paragraph in section 2 to briefly review the recent development in wayfinding technology and explain why the case study is important to this research. The design of the proposed app is all based on the results of behavioral mapping and visibility analysis.